# Computer-Based Cognitive Training in Children with Primary Brain Tumours: A Systematic Review

**DOI:** 10.3390/cancers14163879

**Published:** 2022-08-11

**Authors:** Francesco Sciancalepore, Leonardo Tariciotti, Giulia Remoli, Danilo Menegatti, Andrea Carai, Giuseppe Petruzzellis, Kiersten P. Miller, Francesco Delli Priscoli, Alessandro Giuseppi, Roberto Premuselli, Alberto E. Tozzi, Angela Mastronuzzi, Nicola Vanacore, Eleonora Lacorte, Allena-Mente Study Group

**Affiliations:** 1National Center for Disease Prevention and Health Promotion, Italian National Institute of Health, 00161 Rome, Italy; 2Fondazione IRCCS Cà Granda Ospedale Maggiore Policlinico, Unit of Neurosurgery, 20122 Milan, Italy; 3Department of Neurosurgery, University of Milan, 20122 Milan, Italy; 4Neurology Ward, San Gerardo Hospital, 20900 Monza, Italy; 5Neurology Section, School of Medicine and Surgery, Milan Center for Neuroscience (NeuroMI), University of Milano-Bicocca, 20126 Milan, Italy; 6Department of Computer, Control, and Management Engineering (DIAG), University of Rome “La Sapienza”, 00161 Rome, Italy; 7Neurosurgery Unit, Neuroscience Department, IRCCS Bambino Gesù Children’s Hospital, 00165 Rome, Italy; 8Division of Hematology and Stem Cell Transplantation, Azienda Sanitaria Universitaria Friuli Centrale (ASUFC), 33100 Udine, Italy; 9Multifactorial and Complex Diseases Research Area, IRCCS Bambino Gesù Children’s Hospital, 00165 Rome, Italy; 10Department of Oncology/Hematology, Cell Therapy Gene Therapies and Hemopoietic Transplant, IRCCS Bambino Gesù Children’s Hospital, 00165 Rome, Italy

**Keywords:** brain tumours, paediatric, cognitive, computer-based, neuro-oncology

## Abstract

**Simple Summary:**

Brain tumour survivors are often burdened by late sequelae, especially neurocognitive deficits, ultimately affecting their quality of life. For many years, treatments for neurocognitive impairments have been limited to educational, pharmacological, home-based interventions, or clinic-based cognitive rehabilitation, but these treatment modalities showed several limits. More recently, cognitive rehabilitation through digital tools to increase cognitive performance through exercises and games is spreading in experimental clinical settings. However, since these are innovative interventions, there is a need to further investigate their effects on cognitive outcomes and quality of life for children with brain tumours. Therefore, in this systematic review, we analyse the current evidence and trends regarding computer-based cognitive rehabilitation in paediatric patients diagnosed with, or survivors of, brain tumours. To our knowledge, this is the first systematic review investigating these new approaches to cognitive rehabilitation in children with brain tumours.

**Abstract:**

**Background:** Late neurocognitive sequelae are common among long-term brain tumour survivors, resulting in significantly worse quality of life. Cognitive rehabilitation through specific APP/software for PC/tablets represents an innovative intervention spreading in recent years. In this study, we aim to review the current evidence and trends regarding these innovative approaches. **Methods**: A systematic literature review was performed. Inclusion criteria were: (i) Studies recruiting patients diagnosed with any brain tumour before 21 years of age; (ii) studies assessing the role of digital interventions on cognitive outcomes. Case reports, case series, reviews, letters, conference proceedings, abstracts, and editorials were excluded. **Results:** Overall, nine studies were included; 152 patients (67.8% males) with brain tumours underwent a digital intervention. The mean age at diagnosis and the intervention enrolment ranged from 4.9 to 9.4 years and 11.1 to 13.3 years, respectively. The computer-based software interventions employed were: Cogmed, Captain’s Log, Fast ForWord, and Nintendo Wii. Most of these studies assessed the effects of cognitive training on working memory, attention, and performance in daily living activities. **Conclusions:** The studies suggest that this type of intervention improves cognitive functions, such as working memory, attention, and processing speed. However, some studies revealed only transient positive effects with a significant number of dropouts during follow-up. Trials with greater sample sizes are warranted. Motivating families and children to complete cognitive interventions could significantly improve cognitive outcomes and quality of life.

## 1. Introduction

Cancer is one of the first causes of death from disease in children under 15, but the survival rate for childhood brain tumours (BT) has improved in recent years, facilitated by continued medical advances in diagnosis and treatment [1].

Children treated for BTs frequently experience decreased intellectual function associated with reduced educational, social, and professional performance in young adulthood [1]. Childhood survivors of BTs are at risk of cognitive deficits, both in association with the disease itself (duration, onset, severity) and with the toxic effects of the treatment.

The most common cognitive disorders are visuo-spatial reasoning, motor functioning, attention, working memory, processing speed, and executive functioning [2,3]. In addition, some deficits in socio-cognitive skills have also emerged due to the long-term effects of childhood cancer, such as isolation, difficulty solving social problems, relationship problems, and peer rejection [3]. Poor quality of life in survivors of paediatric BTs was reported by Mostow et al. [4] who examined 342 adults who had been treated for brain tumours before the age of 20 and who had survived at least 5 years. Compared with their siblings, patients exhibited a higher risk for unemployment, chronic health problems, and difficulty driving a motor vehicle [4].

Demographic variables associated with an increased risk of late cognitive and social functioning include the youngest age at diagnosis/treatment, female gender, and cranial radiation therapy. Medical complications linked to diagnosis and treatment can also put survivors at additional risk for late effects, including stroke, seizures, surgical complications, and increased intracranial pressure/hydrocephalus [4].

Moreover, standard adjuvant tumour treatments, such as radiotherapy and chemotherapy, are associated with cognitive impairments. Radiotherapy affects white-matter tracts and the cerebral vasculature, leading to demyelination, vessel walls thickening, focal mineralization, and coagulative necrosis [5]. 

Furthermore, chemotherapy has toxic effects on cognition due to the development of acute and chronic encephalopathy. An animal study showed that temozolomide, an alkylating agent commonly used in glioma treatment, decreased neurogenesis in the hippocampus with a harmful impact on memory encoding and learning [6]. The use of steroid or pain medications, high seizure burden, and several antiepileptic drugs may also exacerbate cognitive problems [6].

For many years, treatments for the neurocognitive sequelae have been limited to pharmacological, educational, home-based interventions, or clinic-based cognitive rehabilitation [7]. However, these interventions showed several limits, such as the reluctance of parents of survivors to pursue special education or pharmacological interventions and poor compliance [8].

More recently, computerised cognitive training programs (CTTP) with or without game-like elements have been experimented to improve neurocognitive functions in survivors of acquired brain injury comprehending patients treated for brain tumours [9]. CTTP collect standardised and challenging tasks targeting specific cognitive domains on the assumption that distinct cognitive abilities might be enhanced by repetitive and demanding performance over time. The previous assumptions have been tested on adults with controversial results and only a few studies reproduced these standardised protocols on children and young adults in experimental clinical settings [9]. Several reviews and meta-analyses on CCTP have been conducted on cognitive training with Cogmed Working Memory Training in patients with neurocognitive disorders. However, most of these works included only/also adult patients, non-computerised cognitive training tools, and no distinction between hospital- and home-based training sections [10,11,12,13]. For these reasons, robust results are still missing and controversial [14]. 

Furthermore, among several reviews and meta-analyses analysing CCTP with or without game-like design, there is a lack of evidence on the effects of CTTP on the cognitive outcomes and quality of life of children with BTs [13]. 

This study aimed to review the current evidence and trends regarding computer-based cognitive rehabilitation in paediatric patients diagnosed with, or survivors of, BTs. 

## 2. Materials and Methods

This systematic literature review was performed according to the methodology described in the Cochrane handbook for systematic reviews [15] and was reported based on the PRISMA statement for reporting systematic reviews and meta-analyses (Table S1) [16]. Additionally, the study protocol was registered in the International Prospective Register for Systematic Reviews (PROSPERO) and was ascribed the CRD42022344520 registration code.

All literature published to July 2021 (updated to June 2022) was retrieved by searching the databases “PubMed”, “Cochrane”, “APA PsycInfo”, and “CINAHL” using the following search terms: **(kinect* OR nintendo* OR “gaming” OR exergam* OR exer-game* OR “virtual reality” OR virtual-reality OR “augmented reality” OR softwar* OR “app” OR tablet* OR smartphone* OR smart-phone* OR *game* OR “machine learning” OR machine-learning OR “artificial intelligence” OR artificial-intelligence OR “deep learning” OR deep-learning OR “convolutional neural” OR convolutional-neural* OR “computer vision” OR computer-vision) AND (child* OR infant* OR adolescent* OR pediatri* OR paediatr* OR poediatr*) AND (cancer* OR tumour* OR neopla* OR malignan* OR *irradiation* OR *proton* OR chemother* OR radiother* OR radio-ther* OR chemo-ther*) AND (cereb* OR brain* OR *crani* OR cns OR spin* OR medull*).**


No limitations in the search strategy were applied to the publication date, study design, or language. References of considered studies were also searched to identify any further relevant data. 

The records identified by the search were uploaded on “Rayyan” [17] to organise the study selection in a more efficient way. The titles and abstracts of the identified records were initially screened and selected by three groups composed of two independent and blinded reviewers (**G.R. + A.C., L.T. + D.M., F.S. + G.P.**) based on their pertinence to the review topic. Conflicts and disagreements were resolved by consensus.

The following set of pre-defined inclusion criteria were then individually applied to the selected articles in their full-text version: (i) Studies recruiting patients diagnosed with any brain tumour before 21 years of age; (ii) assessing the role of digital interventions based on apps, video games, augmented reality, or any other type of software based on cognitive outcomes. Case reports, case series, reviews, letters, conference proceedings, abstracts, and editorials were excluded. Articles not published in English were removed. Systematic reviews were considered separately to check for the consistency of the data. Data extraction was performed by three reviewers (**L.T., G.R. and D.M.**). The extracted data are reported in Table 1.

## 3. Results

Bibliographic searches on literature databases yielded 2033 records. After a first screening, 13 records were selected. Of these, four were further excluded, as they did not meet the inclusion criteria. Overall, nine studies were included. Two of the nine were open-label studies and seven were randomised controlled trials (RCT). All included studies were published in the interval 2011–2021. The flow diagram of included studies is reported in Figure 1. A high agreement (>90%) concerning the inclusion of the records was reported by the reviewers involved in the study selection process (**L.T.; G.R.; D.M.**) and conflicts in the screening process were resolved by inter-personal discussion. The summary of the characteristics of the included studies are reported in Table 1. 

### 3.1. Quality Assessment of the Studies

Quality of the included studies was assessed through the Cochrane Risk of Bias (RoB) tool [18] and reported in Figure 2.

The assessment showed different qualities of the studies: Two studies [19,20] showed a deficient quality, one study low [21], four studies moderate [5,22,23,24], and two trials [25,26] showed high quality, according to Cochrane standards.

The main reasons associated with a higher quality were: The randomisation of participants before the intervention (reported in 7/9 studies, 78%) and a clear blinding of participants and personnel before and during the outcomes assessment (described in 4/9 studies, 44%). On the other hand, the principal reasons associated with a poorer quality were represented by: (i) Incomplete data regarding participants related to significant drop-outs during the follow-up, and (ii) the absence of randomisation of the participants before the digital intervention. 

### 3.2. Population

Overall, 152 patients (67.8% males) with BTs were included and underwent digital intervention. The mean age at diagnosis (not reported in three studies) and at the intervention enrolment ranged from 4.9 to 9.4 years and 11.1 to 13.3 years, respectively. The principal tumour types included were: Medulloblastomas, ependymomas, primate neuroectodermal tumours, astrocytomas, and germ-cell tumours. Five studies also enrolled a control population whose mean age at the intervention ranged from 10.7 to 13.2 years.

In most of the studies, all patients were examined after an average of 5-year intervals at the end of chemo-radiation therapy. In the study by Palmer et al. [22], patients underwent computer-based cognitive training after diagnosis of a primary BT in parallel to appropriate standards of care (SOC), including surgery, cranial and spinal radiation, and/or chemotherapy. In another study [21], authors enrolled patients who underwent cognitive training after surgery (Cases: 12.11 ± 16.97; Controls: 8.72 ± 13.92 months after surgery) in parallel to radiation or medical SOC.

Four studies enrolled mixed populations, including patients diagnosed with primary brain tumours (BT) or acute lymphocytic leukaemia (ALL). When miscellaneous cases were included in intervention studies, the primary justification reported by the authors was the high prevalence of neurocognitive and psychosocial deficits in these populations of patients undergoing chemo-radiation as SOC. However, some authors clarified how patients diagnosed with BTs are more likely to present more extensive and multi-domain cognitive deficits than ALL. Descriptive statistics of mixed populations have been reported, addressing patients diagnosed with BTs only in this section, while all outcomes described are intended as overall findings reported in the original investigations (additional information is available in Table 1).

### 3.3. Cognitive Training Interventions and Outcomes Assessment

All of the studies, except for two [19,20], were designed as randomised controlled investigations, in which a computer-based cognitive training was assigned to intervention or control groups. In four studies [5,22,23,26], the participants were randomised to experimental cognitive training, according to the study protocol or the waiting list group. In comparison, in three studies [21,24,25], the enrolled patients were assigned to adaptive- or not-adaptive computer-based training. The latter differed for active or absent progressive difficulty adjustment during cognitive tasks. Drop-out was higher in intervention programs, especially in groups undergoing adaptive computer-based training.

The intervention computer-based software programs employed included: **Cogmed** [27] (n = 6; https://www.cogmed.com, accessed on 12 January 2022), **Captain’s Log** [28] (n = 1; https://www.braintrain.com, accessed on 12 January 2022), **Fast ForWord** [29] (n = 1; https://www.scilearn.com, accessed on 12 January 2022), and **Nintendo Wii** [30] (n = 1; https://www.nintendo.com, accessed on 12 January 2022). For extended information regarding the computer-based interventions, see Appendix A.

The majority of studies investigated the effects of cognitive training on working memory, attention, and performance in daily living activities (with peers and family). The primary outcome measures were: Wechsler Intelligence Scale for Children, Fourth edition [31], Working Memory Index (WMI), National Institute of Health Toolbox Cognition Battery (NTCB) [32], Conners’ Parent Rating Scale (CPRS) [33], and a customised feasibility questionnaire. The intervention duration varied from 5 to 12 weeks among studies. All of the enrolled patients underwent baseline and post-intervention, or control cognitive assessment with a variable follow-up interval (3 months to 5 years).

### 3.4. Qualitative Summary of the Studies

Most of the studies (6/9, 67%) trained participants with BTs through the administration of Cogmed. Cogmed represents digital training for improving working memory and attention. In the included studies, the training period ranged from 5 to 12 weeks. All of the studies, except for one [20], asked participants to complete 25 at-home training sessions, while Carlson-Green et al. [20] reported 35 training sessions.

Five of six studies were RCTs. Three of them [21,24,25] randomly assigned participants to two different groups: Adaptive vs. not-adaptive Cogmed versions. In the adaptive version, the difficulty of the tasks was adjusted throughout each of the 25 training sessions, in order that as the child became more skilled, the exercises became more difficult. Conversely, in the not-adaptive condition, the computer program consisted of the same activities, with the level of difficulty that never increased. Therefore, the not-adaptive condition was intended to provide a non-therapeutic “dose” of the intervention.

Overall, results reported significant improvement in visual working memory skills in those participants who were compliant with the adaptive training program compared with those who completed the not-adaptive version. In the study by Siciliano et al. [21], the efficacy of the interventions diminished over time on WISC-IV WMI [F (2, 28) = 3.28, *p* = 0.05] and the NTCB Fluid Cognition Composite scores [F (3, 33) = 8.45, *p* < 0.001]. However, in this latter study there was no evidence of stable improvement over time, thus favouring the adaptive to the not-adaptive version of Cogmed. Furthermore, in one study [25], training had no significant effects on attention or verbal working memory tasks. Differences in parent ratings between the adaptive and not-adaptive groups were no longer significant at the 3-month follow-up evaluation [F = 3.65, *p* = 0.08].

Two studies [5,23] randomised participants to intervention (n = 30) vs. waiting list (n = 32). In one study [5], authors reported that the intervention group demonstrated greater short-term improvement than the control group (*p* = 0.02) for the primary outcome measure (spatial span backward). Moreover, the intervention group demonstrated greater short-term improvement than the control group on secondary measures of attention (*p* = 0.01), working memory (*p* = 0.02), and processing speed (*p* = 0.02). In the other study [23], neither the control nor the intervention group showed acute or long-term decline in peer relation problems through the 6-month post-assessment. The waitlist group exhibited significantly elevated self-reported problems at baseline and showed an acute decline in family relation problems at immediate post-waitlist assessment, while the intervention group reported no significant acute change. Neither group experienced a decline in family relations over the 6 months.

Moreover, the intervention group significantly showed more remarkable improvement on several measures of working memory (*p* < 0.05) and attention (*p* < 0.01) than the control group. Parents of participants in the intervention group described more significant reductions in attention and executive functioning problems than parents of controls (Conners 3 Inattention and Executive Function Scales, *p* < 0.05). However, the authors admitted that parents’ reports might have been affected by observer bias in investigation vs. control and adaptive-training vs. not-adaptive training version by guessing the intervention type.

The study by Palmer et al. [22] employed the intervention “Fast ForWord” to improve reading ability in patients (n = 81) with BTs. In addition to receiving the SOC, patients were randomly assigned to the intervention group. Seventeen of the 43 patients (39.5%) randomised to the intervention group were able to complete the target of 30 intervention sessions. Authors reported no change in reading ability over time (5 years) between control and intervention group (*p* = 0.62).

In another trial, Sabel et al. [26] showed that motor and process skills of participants (n = 7) improved significantly after active video gaming (Nintendo Wii). However, authors did not report significant changes in cognitive performances (general working memory, verbal and visuo-spatial learning) although trends for improvement in sustained attention and selective attention were reported.

The long-term results of the approaches hereby discussed are still controversial. Most of the authors reported that further studies are mandatory to explore the effects on additional cognitive domains other than working memory and the duration of improvement or reduction of cognitive decline in patients undergoing SOC for BTs, or those who are BT survivors and had previously undergone radiation- and chemo-therapies. Extended data are reported in Table 1.

## 4. Discussion

In this systematic review, we summarised the available evidence of the effects of computer-based interventions on cognitive outcomes in patients diagnosed with, or survivors of, paediatric BTs.

Overall, the studies suggest that this intervention improves cognitive functions, such as working memory and attention. Specifically, in those studies characterised by intervention group vs. waiting list [5,23], participants in the intervention group exhibited more significant cognitive improvements than the control group.

Furthermore, in two studies [21,25], the adaptive version of the training (Cogmed) led to higher improvements in working memory skills compared with participants who completed the not-adaptive version of the program. These results indicate that computerised cognitive training may represent a productive alternative to pre-existing interventions, contributing to a significant advancement in the management of cognitive functions. However, some trials depicted a scenario in which there was no significant improvement in cognitive tests over time. Several factors may explain these different results.

The first important variable to be considered is represented by the type of intervention employed in the studies. Cogmed has led to more significant results in improving cognitive functions (working memory and attention) compared with the other digital interventions (Nintendo Wii, Captain’s Log, and Fast ForWord). Cogmed showed significant advantages compared with other computerised cognitive training programs for the survivor population. Its use has been associated with significant efficacy in a number of controlled trials with children and adolescents affected by ADHD and other pathologies associated with poorer performances in the domains of working memory and attention [34,35,36]. Moreover, Cogmed specifically targets working memory skills, which have been proposed to underlie the changes in intelligence and academic performance frequently seen in survivors with cognitive late effects [37]. Finally, this computer-based training consists of a fixed “dose” of training (i.e., 25 sessions) and an active control version of the program, making it ideally suited for empirical study in a randomised controlled design.

Another important factor associated with the different results which emerged in the studies may be represented by the timing of the intervention. Specifically, most of the studies enrolled patients after a mean of 5 years post-SOC. However, in some studies [21,22], patients were enrolled in parallel with the radiation therapy or chemotherapy, showing a lower completion of the digital intervention. In this regard, most of the participants were not able to complete the predefined “acceptable dose” of Cogmed sessions, indicating that patients were potentially overburdened by study procedures [21]. On the other hand, those studies conducting interventions after a minimum of 2 years, and on average 5 years after completion of all treatment, when completing an at-home online training program showed more feasibility and acceptance. These findings highlight that the delivery of interventions closer to diagnosis and treatment may be more stressful for families leading to higher number of dropouts and a lower efficacy of the intervention.

Feasibility and acceptance represent other essential variables to take into account. Evaluation of feasibility and acceptance plays a crucial role in validating computer-based interventions. Indeed, in those studies reporting higher rates of these two variables, there were significant improvements in cognitive and social outcomes compared with those studies indicating a poorer feasibility of the intervention. The study by Hocking et al. [24] investigated the acceptance and feasibility of Cogmed, showing significant feasibility difficulties in using the intervention, in which only half of the sample completed at least 20 sessions. Despite incentives for completing sessions, 18.5% of the sample did not conduct any training sessions and 48.1% did not complete the intervention, indicating a failure to engage paediatric patients with BTs in Cogmed. These findings contrast with other studies employing Cogmed that showed a good feasibility. To this regard, the contrasting results may be due to methodological differences, as the other studies [5,23,25] were largely also comprised of ALL survivors who may have less severe neurocognitive deficits than patients with BTs, thus being able to complete a higher number of training sessions. Overall, most of the trials reported a good degree of feasibility and acceptance in participants with BTs, despite the three studies [21,22,24] showing significant dropouts during the follow-up.

In addition to the aforementioned differences, it should be emphasised that the included studies enrolled small sample sizes (5/9 had a number of participants < 30). For this reason, despite reporting some significant results, most of the studies were not strong enough to evaluate the efficacy of these interventions. Moreover, the limitation of the samples is due to the fact that BTs in paediatric age represent a rare pathology and the recruitment of patients is challenging. In this regard, few studies assessed cognitive digital interventions in the paediatric population with BTs. In this review, only four studies evaluated the computer-based intervention exclusively in a population with BTs; one was not an RCT, thus representing a limitation regarding the quality of the results. To achieve more encouraging results, a higher number of trials with larger sample sizes will be needed.

As mentioned above, nearly 80% of the included studies assessed patients after the end of the standard treatments for BTs. The mean age of participants at the time of the assessment was 12.24 ± 0.59 years, with a mean of 7 years after the diagnosis (5.82 ± 1.49 years). Although children diagnosed with a BT can experience significant cognitive effects years after treatment, there is a need to start therapies closer to the diagnosis. Recent evidence suggests that deficits may emerge soon after or even prior to surgery, indicating the potential importance of delivering interventions as early as possible [38]. Therefore, early interventions could lead to a lower risk for subsequent cognitive decline and social impairments. However, the potential difficulties due to lower feasibility of computer-based interventions during the first phases after the diagnosis, as reported in the study by Siciliano et al. [21], represent a problem to be addressed. Utilization of at-home computerised training may be challenging for children while still actively dealing with the stress of their diagnosis. Accurate psychoeducation and understanding of cognitive late effects may be valuable in motivating families to complete the interventions [39]. In addition, strong collaboration between psychologists and medical providers will be beneficial in encouraging families and children to complete cognitive interventions, regardless of the timing of when they are delivered [21].

Despite some limitations and differences across all the studies, evidence from this review supports the idea that computer-based interventions are promising in cognitive rehabilitation. These results show similarities to evidence outlined in a systematic review [40] that assessed the efficacy of remote technology-based training programs (TP) for children with acquired brain injury (ABI). Based on the review process, 16 of the 18 studies on cognitive TP were found to be effective in improving cognitive outcomes immediately after training, on both near- and far-transfer measures. Moreover, 8 of the 12 studies on cognitive TP that considered also outcomes on functioning in daily life found positive gains. Moreover, another systematic review regarding cognitive rehabilitation on children and adolescents with ABI suggested that these types of interventions are promising, as they can lead to improvements in both cognitive and psychosocial functioning of children with ABI [41].

However, it is important to outline, that digital interventions should always follow crucial steps to achieve their aims. First, a study on the usability/feasibility of the digital tool should be conducted. Then, a trial with an adequate sample size should be designed to assess the device’s safety and efficacy. By following these fundamental phases, the intervention may improve to reach final approval to become a digital therapy. Digital therapeutics are not simple applications concerning health, nor simple systems offered by pharmaceutical companies that help patients manage their diseases, instead they represent real curative interventions, enabling the improvement of clinical outcomes, such as a pharmacological treatment [42]. In 2012, the European Commission released guidance (updated in 2016) on the qualification and classification of stand-alone software used in the health care setting as a medical device [43]. Under this guidance, “mobile apps are considered medical devices if used specifically for diagnostic and/or therapeutic purposes, including the diagnosis, prevention, monitoring, treatment, or alleviation of disease”.

There are significant differences in the structure and organization regarding medical products and software registration between the European authorities and the US Food and Drug Administration (FDA). The EMA and the FDA are responsible for pharmaceutical regulation, but only the FDA has the control on pharmaceuticals and medical devices. In the EU, only the European Commission (no single agency) is responsible for regulating digital therapeutics [43]. To this regard, a clear set of regulations should be defined at national and international levels to support interoperability and allow for safe and effective data exchange between different information and communication technology systems [42].

By improving these clinical and normative aspects, digital therapeutics could be safer and more effective for patients. It will indeed lead to an improvement of the quality of life of the patients themselves and their families.

## 5. Conclusions

This paper aimed to systematically review the evidence of computer-based cognitive interventions in paediatric patients diagnosed with BTs. Overall, the included studies showed a positive trend of these interventions on cognitive outcomes. Working memory and attention were the most trained and improved cognitive functions by the digital treatments. Nevertheless, some studies revealed only transient positive effects on these neurocognitive domains with a significant number of dropouts during the follow-up. More trials with larger sample sizes are warranted. Moreover, there is a need for trials to enrol patients closer to the time of diagnosis, even though patients and their families may experience a poorer feasibility and acceptance of the treatment. To this regard, strong collaboration between medical providers and psychologists could be beneficial in motivating families and children to complete cognitive interventions, thus leading to potentially significant improvements of cognitive outcomes and quality of life.

## Figures and Tables

**Figure 1 cancers-14-03879-f001:**
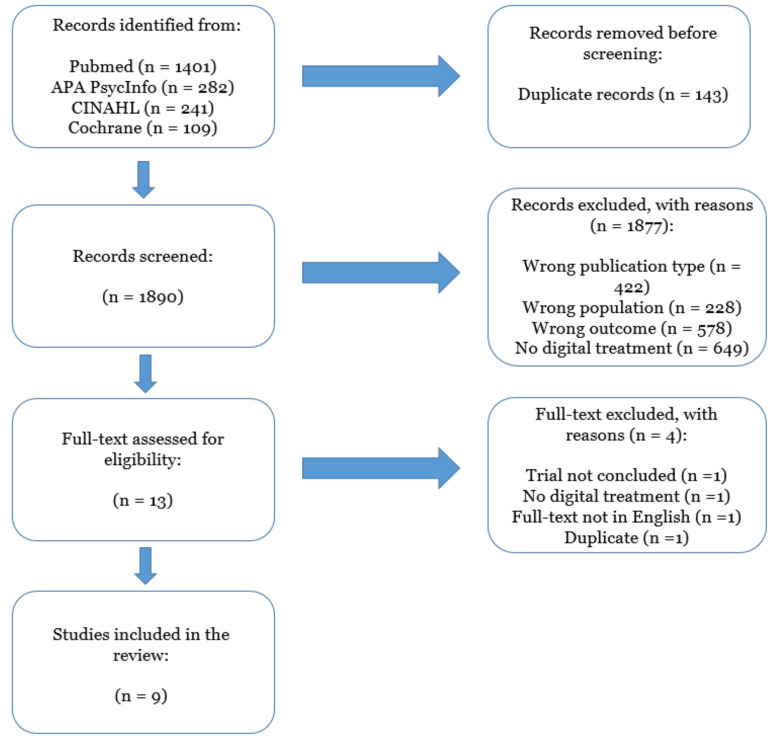
PRISMA flow chart describing the inclusion process of the articles.

**Figure 2 cancers-14-03879-f002:**
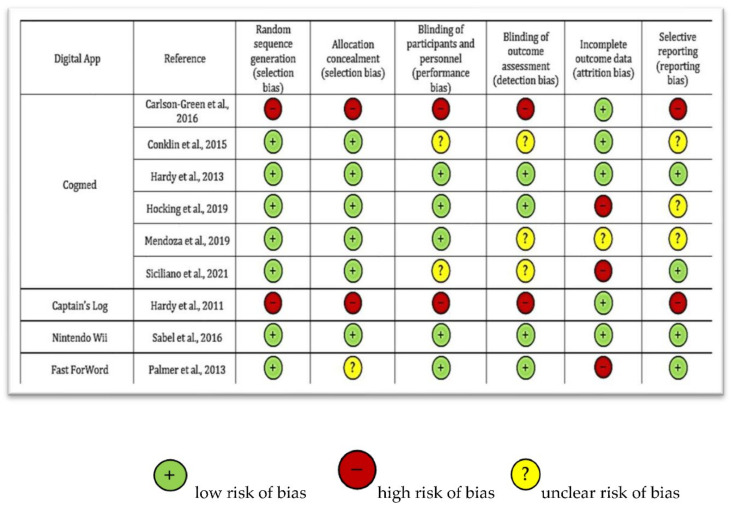
Qualitative assessment of the included studies through Cochrane Risk of Bias tool [5,18,19,20,21,22,23,24,25,26].

**Table 1 cancers-14-03879-t001:** Main data of the included studies.

Author	Year	No. of Participants (Cases and Controls)	Gender (M, F%) and Mean Age	Mean Age at Diagnosis (Years)	Type of Neoplastic Disease	Intervention	Cognitive Abilities Trained	Attrition	Results (Summary of Findings)
Carlos-Green	2016	N = 21 (21 intervention) No control group.	NR; age range 8 to 18 years.	6 (1–14)	Medulloblastoma (N = 11), Germinoma (N = 4), Ependymoma (N = 4), and other tumour types (N = 2).	Name: Cogmed. Participants in this study were asked to complete 35 training sessions over 8 to 12 weeks and were contacted by telephone to check progress and enhance motivation. At the end of the training, participants and their parents also completed questionnaires assessing their satisfaction with the program. Participants were assessed through a follow-up testing 6 months after the training completion.	Attention;Working memory; Executive functions.	2/21 (9.5%) participants did not complete all the training sessions.	The efficacy of Cogmed was examined 6 months after completing the intervention. Improvements: Working memory (verbal and visual-spatial tasks), academic math test, executive functioning (emotional and behavioural control, ability to transition/shift between activities, planning and organizational skills, ability to monitor their behaviour).
Conklin	2015	N = 68. Intervention (N = 34); Control group waiting list (N = 34).	Cases: 18 M (52.9%) 16 F (48.1%), age 12.2 ± 2.5; Controls: 18 M (52.9%) 16 F (48.1%), age 11.8 ± 2.4.	Cases: 5.2 ± 2.9; Controls: 4.6 ± 2.7.	Cases: 23 ALL, 11 BT (8 Medulloblastoma, 2 Glioma, 1 Ependymoma); Controls: 24 ALL, 10 BT (7 Medulloblastoma, 3 Ependymoma).	Name: Cogmed. Participants were randomly assigned to the intervention. The intervention group was asked to complete 25 at-home training sessions over 5 to 9 weeks. Training progress was monitored over the Internet and coaching telephone calls were used to provide feedback and help maintain motivation. All participants had a final cognitive assessment after 6 months.	Attention; Working memory; Executive functions.	Cogmed group: 4 participants (11.8%) incomplete trainings. 30 follow-up assessments. Controls: 2 dropouts (5.9%). 32 follow-up assessments.	Improvements in the intervention group: Spatial span backward short-term (*p* = 0.002); WISC-IV spatial span forward (*p* = 0.012); CPT-II omissions (*p* = 0.036), WM (WISC-IV digit span backward, *p* = 0.017; WISC-IV working memory index, *p* = 0.022), and processing speed (CPT-II reaction time, *p* = 0.020). Improvements also regarding attention and executive functions compared with the control group participants (CPRS-3 inattention, *p* = 0.009; CPRS-3 executive function, *p* = 0.002).
Hardy	2011	N = 9 (9 intervention) No control group.	Cases: 5 M (56%) 4 F (44%), age 13.3 ± 2.4.	NR	Cases: 3 ALL, 6 BT (1 Primitive neuroectodermal tumour, 3 Medulloblastoma, 2 Ependymoma).	Name: Captain’s Log. Participants were asked to complete a 50 min/week training session for 12 weeks. 3 months after the completion, participants returned to the clinic for follow-up testing.	Problem solving; Working memory; Attention.	1 participant (11.1%) did not complete all follow-up visits.	Working memory scores increased from baseline to the follow-up assessment [F(2,15.11) = 3.16; *p* = 0.07]. Digit span forward had a significant increase over time [F(2,15.09) = 6.79; *p* < 0.01)]. Attention problems [F(2,15.10) = 6.98; *p* < 0.01], significantly decreased across 3 time points. Digit span backward [F(2,15.27) = 0.10; NS] and number sequencing [F(2,15.38) = 0.40; NS] did not improve significantly post-intervention.
Hardy	2013	N = 20. Intervention (N = 13); Active control group (N = 7; training: Not-adaptive Cogmed).	Cases: 8 M (61.5%) 5 F (38.5%), age 12.7 ± 2.77; Controls: 4 M (57.1%) 3 F (42.9%), age 10.7 ± 1.89.	Cases: 4.9 ± 3.54; Controls: 5.7 ± 2.88.	Cases: 7 ALL, 6 BT (2 Medulloblastoma/PNET, 3 Ependymoma, 1 other tumour type); Controls: 4 ALL, 3 BT (2 Medulloblastoma/PNET, 1 other tumour type)	Name: Cogmed. Participants were randomly assigned to the success-adapted computer intervention or not-adaptive active control condition. Participants were asked to complete 25 at-home training sessions (3 to 5 sessions a week) over 5 to 8 weeks. Participants were assisted by a treatment coach to motivate them and solve problems. Follow-up assessment after 3 months.	Attention; Working memory; Executive functions.	Cases: 2 incomplete trainings (15.4%); Controls: 1 incomplete training (14.3%).	Symbolic working memory task from the WRAML2—the cases increased significantly [F = 4.57, *p* = 0.05] compared with the controls during the intervention period, while this effect was no longer significant at the 3-month follow-up [F = 3.65, *p* = 0.08]. Cases experienced a greater improvement in parent-reported learning problems on the Conner-3 [F = 4.65, *p* = 0.05]. Moreover, 45% of cases exhibited improvement consistent with the RCI, even after the 3-month follow-up.
Hocking	2019	N = 27. Standard intervention (N = 14); Active control group (N = 13; training: Cogmed + Parent intervention).	14 M (51.9%) 13 F (48.1%), mean age 11.07 (7–16).	Cases: 4.96 ± 3.48.	7 Astrocytoma, 6 Medulloblastoma, 6 Ependymoma, 1 low-grade glioma, 7 other BTs.	Name: Cogmed. Participants in both groups were assigned to 25 computer sessions over 5 to 6 weeks (30–45 min for each session). Participants in the combined intervention were also exposed to a “Parent intervention”: Phone sessions for parents in the combined group included six sessions (duration: 30–45 min) regarding manualised problem-solving skills training (PSST).	Attention; Working memory; Executive functions.	5 participants (18.5%) lost to follow-up in both standard and combined group. In the next 3 months, standard group lost 3 further participants.	Completers: Better performance in baseline auditory attention abilities (digit span forward) than non-completers and they also showed a reduction of working memory difficulties in completers than non-completers.
Mendoza	2019	N = 68. Intervention (N = 34); Control group waiting list (N = 34).	Cases: 18 M (53%) 16 F (47%), age 12.21 ± 2.47; Controls: 18 M (53%) 16 F (47%), age 11.82 ± 2.42.	Cases: 5.15 ± 2.92; Controls: 4.62 ± 2.68.	Cases: 23 ALL, 11 BT (8 Medulloblastoma, 2 Glioma, 1 Ependymoma); Controls: 24 ALL, 10 BT (7 Medulloblastoma, 3 Ependymoma).	Name: Cogmed. Participants were randomly assigned to computerised training or waitlist control groups. Participants in the Cogmed intervention group were asked to complete 25 at-home training sessions over 5 to 9 weeks. The exercises increased or decreased in difficulty and complexity based on performance. Progress and participants’ motivation were monitored by coaching phone calls.	Attention; Working memory; Executive functions.	Cogmed group: 4 incomplete trainings (11.8%). 30 follow-up assessments. Controls: 2 drop-outs (5.9%). 32 follow-up assessments.	From baseline to post-intervention assessment, the intervention group showed greater improvement than the control group on: Attention and working memory (WISC-IV spatial span forward, digit span backward, working memory index, *p* < 0.05; WISC-IV spatial span backward *p* < 0.001). Improvements also in executive functioning and attention.
Palmer	2013	N = 81. Intervention (N = 43); Control group waiting list (N = 38).	Cases: 24 M (55.8%) 19 F (44.2%), age NR; Controls: 26 M (68.4%) 12 F (31.6%), age NR.	Cases: 9.38 ± 3.12; Controls: 9.27 ± 3.18.	Cases: 43 Medulloblastoma; Controls: 38 Medulloblastoma.	Name: Fast ForWord. In addition to the standard-of-care, patients were asked to complete the Fast ForWord computer-based training program 48 min/day, 5 days/week, for 6 weeks—30 sessions, with a total training time of 1440 min. Participants were assisted by a teacher and their performance was monitored. 5-year follow-up period.	Working memory; Attention; Auditory processing and sequencing; Reading ability.	Cases: 3 incomplete trainings (6.9%); 2 incomplete assessments (4.7%).	Patients with high-risk disease (*p* = 0.0042) and younger age at diagnosis (*p* < 0.0001) had more declines in reading during the follow-up. Older patients at diagnosis date had less decline in reading (*p* = 0.0008) and decoding (*p* = 0.0367).
Sabel	2016	N = 13. Intervention (N = 7); Control group waiting list (N = 6).	Cases: 3 M (43%) 4 F (57%), age 11.9 ± 3.6; Controls: 3 M (50%) 3 F (50%), age 13.2 ± 1.9.	NR	Cases: 1 Anaplastic Astrocytoma, 2 Germinoma, 1 Medulloblastoma, 2 Pilocytic Astrocytoma, 1 Supratentorial Primitive Neuroectodermal tumour; Controls: 1 Choroid Plexus Carcinoma, 1 Germinoma, 2 Medulloblastoma, 1 Pilocytic Astrocytoma, 1 Supratentorial Primitive Neuroectodermal tumour.	Name: Nintendo Wii, Wii-Fit. Patients were randomly assigned to the intervention. Participants were asked to complete a minimum of 30 min/day, at least 5 days/week, over 10 to 12 weeks. Activity levels were measured via a multisensory activity monitor for 1 week at baseline, every second week during the intervention period, and for 1 week after the waiting list period.	Body coordination; Hand-eye coordination; Fine motor control.	No attrition found.	The intervention group exhibited improvement in: Motor (*p* = 0.012) and process (*p* = 0.002) skills after active video gaming. There were no significant changes in cognitive tests, although positive trends in selective (*p* = 0.078) and sustained attention (*p* = 0.090).
Siciliano	2021	N = 41. Intervention (N = 20); Active control group (N = 21; training: Not-adaptive Cogmed).	Cases: 13 M (65%) 7 F (35%), age 12.31 ± 2.57; Controls: 13 M (57%) 7 F (43%), age 11.67 ± 2.81.	NR	NR	Name: Cogmed. Participants were randomly assigned to adaptive or not-adaptive versions. Participants were asked to complete 25 sessions (30–45 min) for 5 days a week, over a 5-week period. Coaches supported participants one to two times per week. Follow-up: 10 to 20 weeks post-intervention, and the final one 6 months after the previous assessment.	Attention; Working memory; Executive functions.	15/41 participants (36.6%) did not complete T2 assessment. The T3 and T4 assessment completion did not vary by group.	WMI and NTCB scores significantly improved immediately post-intervention compared with baseline scores. No significant differences between adaptive and not-adaptive conditions.

**Abbreviations used:** NR: Not reported; BT: Brain tumour; ALL: Acute Lymphoblastic Leukemia; WM: Working Memory; WISC: Weschler Intelligence Scale for Children; CPT: Conners Continuous Performance Test; NTCB: National Institute of Health Toolbox Cognition Battery; CPRS: Conners’ Parent Rating Scale; WRAML2: Wide Range Assessment of Memory and Learning 2; WMI: Working Memory Index.

## Data Availability

The data presented in this study are available in the article “Computer-based cognitive training in children with primary brain tumours: a systematic review”.

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
