# Peer review of "Computer-Based Cognitive Training in Children with Primary Brain Tumours: A Systematic Review"

_cancers, 2022, doi:10.3390/cancers14163879_

Round 1

Reviewer 1 Report

Comments to the Authors

The paper presents a systematic review on computer-based cognitive training for children with primary brain tumours.

I report below some suggestions to improve the manuscript quality.

Introduction

Authors should consider to include more references, also more recent. For example, the reference related to the cognitive disorders of children with brain tumor is only one (ref. 2) and is also outdated (2007). For lines 78-80 no reference is reported.

Authors should also report and briefly discuss the recent findings of the literature on remote cognitive training for pediatric populations having a brain injury. To do this, citing reviews instead of single studies would provide a more complete overview. Authors should address the following questions: Which is the state of the art on the topic? What are the main findings of studies and the abilities more susceptible to change? Are there clinical or cognitive characteristics of patients that seem to be more easily associated with benefits?

Some previous reviews on cognitive training for children with brain injury or neuropsychological disorder are outlined below:

- Linden M, Hawley C, Blackwood B, Evans J, Anderson V, O'Rourke C. Technological aids for the rehabilitation of memory and executive functioning in children and adolescents with acquired brain injury. Cochrane Database Syst Rev. 2016 Jul 1;7(7):CD011020.

- Corti C, Oldrati V, Storm F, Bardoni A, Strazzer S, Romaniello R. Remote cognitive training for children with congenital brain malformation or genetic syndrome: a scoping review. J Intellect Disabil. 2022 May 18:17446295221095712.

- Oldrati V, Corti C, Poggi G, Borgatti R, Urgesi C, Bardoni A. Effectiveness of Computerized Cognitive Training Programs (CCTP) with Game-like Features in Children with or without Neuropsychological Disorders: a Meta-Analytic Investigation. Neuropsychol Rev. 2020 Mar;30(1):126-141.

- Corti C, Oldrati V, Oprandi MC, Ferrari E, Poggi G, Borgatti R, Urgesi C, Bardoni A. Remote Technology-Based Training Programs for Children with Acquired Brain Injury: A Systematic Review and a Meta-Analytic Exploration. Behav Neurol. 2019 Aug 1;2019:1346987.

- Lindsay S, Hartman LR, Reed N, Gan C, Thomson N, Solomon B. A Systematic Review of Hospital-to-School Reintegration Interventions for Children and Youth with Acquired Brain Injury. PLoS One. 2015 Apr 29;10(4):e0124679.

- Resch C, Rosema S, Hurks P, de Kloet A, van Heugten C. Searching for effective components of cognitive rehabilitation for children and adolescents with acquired brain injury: A systematic review. Brain Inj. 2018;32(6):679-692.

Material and Methods:

Line 138: “Data extraction was performed by three reviewers”. Please report which data have been extracted for each study.

Results:

Line 146: “A high agreement (>90%) was reported by the reviewers involved in the study selection process”. Could the Authors better explain this point? What is the agreement referred to? How have they calculated it?

Tables

Table 1: it could be useful to insert the cognitive abilities trained by each program and, in the “Results” column, to insert both the domains for which an improvement was found and those for which no benefits were detected. In addition, in the column “Attrition” data would be clearer if reported as percentages.

Further, the type of control group (waiting-list, active –offered with what kind of intervention-), if any, should be better reported. Authors could consider to report data in the following way:

Control group: no control group

                       waiting-list (N=…)

                        active control group (N=…; training:…)

                        etc…

Figures

Figure 1: in the cell reporting study exclusion, reasons for exclusion should be listed.

Discussion

The Authors could discuss their results also taking into account findings of previous studies on remote cognitive training in clinical pediatric samples (see the studies mentioned above) in order to provide an overview that suggests a clear knowledge on the topic. Could there be differences in terms of training success in brain tumor patients as compared to other studied populations? This is an important aspect to discuss to contribute to both research knowledge and clinical practice.

References

As already stated, bibliography could be improved by adding more references. Now only 36 references are reported, please consider to add more studies, also more recent.

Author Response

Reviewer 1:

Introduction

Authors should consider to include more references, also more recent. For example, the reference related to the cognitive disorders of children with brain tumor is only one (ref. 2) and is also outdated (2007). For lines 78-80 no reference is reported.

Authors should also report and briefly discuss the recent findings of the literature on remote cognitive training for pediatric populations having a brain injury. To do this, citing reviews instead of single studies would provide a more complete overview. Authors should address the following questions: Which is the state of the art on the topic? What are the main findings of studies and the abilities more susceptible to change? Are there clinical or cognitive characteristics of patients that seem to be more easily associated with benefits?

Some previous reviews on cognitive training for children with brain injury or neuropsychological disorder are outlined below:

- Linden M, Hawley C, Blackwood B, Evans J, Anderson V, O'Rourke C. Technological aids for the rehabilitation of memory and executive functioning in children and adolescents with acquired brain injury. Cochrane Database Syst Rev. 2016 Jul 1;7(7):CD011020.

- Corti C, Oldrati V, Storm F, Bardoni A, Strazzer S, Romaniello R. Remote cognitive training for children with congenital brain malformation or genetic syndrome: a scoping review. J Intellect Disabil. 2022 May 18:17446295221095712.

- Oldrati V, Corti C, Poggi G, Borgatti R, Urgesi C, Bardoni A. Effectiveness of Computerized Cognitive Training Programs (CCTP) with Game-like Features in Children with or without Neuropsychological Disorders: a Meta-Analytic Investigation. Neuropsychol Rev. 2020 Mar;30(1):126-141.

- Corti C, Oldrati V, Oprandi MC, Ferrari E, Poggi G, Borgatti R, Urgesi C, Bardoni A. Remote Technology-Based Training Programs for Children with Acquired Brain Injury: A Systematic Review and a Meta-Analytic Exploration. Behav Neurol. 2019 Aug 1;2019:1346987.

- Lindsay S, Hartman LR, Reed N, Gan C, Thomson N, Solomon B. A Systematic Review of Hospital-to-School Reintegration Interventions for Children and Youth with Acquired Brain Injury. PLoS One. 2015 Apr 29;10(4):e0124679.

- Resch C, Rosema S, Hurks P, de Kloet A, van Heugten C. Searching for effective components of cognitive rehabilitation for children and adolescents with acquired brain injury: A systematic review. Brain Inj. 2018;32(6):679-692.”

Response:

Thank you for your comments and suggestions. We have included more references (from 36 to 43) and we have cited some recent reviews in the introduction to provide a more complete overview.

Reviewer 1:

“Material and Methods:

Line 138: “Data extraction was performed by three reviewers”. Please report which data have been extracted for each study.”

Response:

We added the reference to the Table 1 which contains the extracted data.

Reviewer 1:

“Results:

Line 146: “A high agreement (>90%) was reported by the reviewers involved in the study selection process”. Could the Authors better explain this point? What is the agreement referred to? How have they calculated it?”

Response:

Thank you for the question. We added in the text that the agreement was referred to the inclusion of the records. We have calculated the percentage (90%) through “Rayyan”, the software we have used to perform this review. Rayyan provided us the percentages of agreement between the reviewers.

Reviewer 1:

Tables

Table 1: it could be useful to insert the cognitive abilities trained by each program and, in the “Results” column, to insert both the domains for which an improvement was found and those for which no benefits were detected. In addition, in the column “Attrition” data would be clearer if reported as percentages.

Further, the type of control group (waiting-list, active –offered with what kind of intervention-), if any, should be better reported. Authors could consider to report data in the following way:

Control group: no control group

                       waiting-list (N=…)

                        active control group (N=…; training:…)

                        etc…”

Response:

Thank you for your suggestions. We have modified the Table 1 following your suggestions.

Reviewer 1:

“Figures

Figure 1: in the cell reporting study exclusion, reasons for exclusion should be listed.”

Response:

Thank you. We have reported the reasons in the figure 1.

Reviewer 1: 

“Discussion

The Authors could discuss their results also taking into account findings of previous studies on remote cognitive training in clinical pediatric samples (see the studies mentioned above) in order to provide an overview that suggests a clear knowledge on the topic. Could there be differences in terms of training success in brain tumor patients as compared to other studied populations? This is an important aspect to discuss to contribute to both research knowledge and clinical practice.”

Response:

Thank you for your advice. We have included in the discussion two recent systematic reviews about the utilization of cognitive training programs on children with Acquired Brain Injury (ABI) and we have reported the results of these reviews, comparing them with our work.

Reviewer 1:

“References

As already stated, bibliography could be improved by adding more references. Now only 36 references are reported, please consider to add more studies, also more recent.”

Response:

Done.

Reviewer 2 Report

The authors present a well conceived  and well-written paper describing a literature review of computer-based cognitive training in children with primary brain tumors. The paper is very well done but a few questions about some areas will help improve clarity:

Introduction:

Are there more recent citations than what is described in the current introduction?

Discussion: 

Overall the authors do a very nice job of outlining important issues learned in the literature. There are recent articles about the impact of screen time on children and on children with brain injury/TBI that would be an important point to consider. Can you include mention of this in the discussion? 

Author Response

Reviewer 2:

“The authors present a well conceived  and well-written paper describing a literature review of computer-based cognitive training in children with primary brain tumors. The paper is very well done but a few questions about some areas will help improve clarity:

Introduction:

Are there more recent citations than what is described in the current introduction?”

Response:

Thank you. We have cited some recent reviews in the introduction to provide a more complete overview.

Reviewer 2:

“Discussion: 

Overall the authors do a very nice job of outlining important issues learned in the literature. There are recent articles about the impact of screen time on children and on children with brain injury/TBI that would be an important point to consider. Can you include mention of this in the discussion?”

Response:

Thank you for your advice. We have included in the discussion two recent systematic reviews about the utilization of cognitive training programs on children with Acquired Brain Injury (ABI) and we have reported the results of these reviews, comparing them with our work.

Round 2

Reviewer 1 Report

The Authors adequately addressed all my comments.